# Factors Associated with SARS-CoV-2 Infection in Physician Trainees in New York City during the First COVID-19 Wave

**DOI:** 10.3390/ijerph18105274

**Published:** 2021-05-15

**Authors:** Kate R. Pawloski, Betty Kolod, Rabeea F. Khan, Vishal Midya, Tania Chen, Adeyemi Oduwole, Bernard Camins, Elena Colicino, I. Michael Leitman, Ismail Nabeel, Kristin Oliver, Damaskini Valvi

**Affiliations:** 1Department of Environmental Medicine and Public Health, Icahn School of Medicine at Mount Sinai, New York, NY 10029, USA; kate.pawloski@mssm.edu (K.R.P.); betty.kolod@mssm.edu (B.K.); rabeea.khan@mssm.edu (R.F.K.); Vishal.midya@mssm.edu (V.M.); tania.chen@icahn.mssm.edu (T.C.); Adeyemi.oduwole@icahn.mssm.edu (A.O.); Elena.colicino@mssm.edu (E.C.); ismail.nabeel@mssm.edu (I.N.); kristin.oliver@mssm.edu (K.O.); 2Department of Surgery, Icahn School of Medicine at Mount Sinai, New York, NY 10029, USA; michael.leitman@mssm.edu; 3Department of Medicine, Icahn School of Medicine at Mount Sinai, New York, NY 10029, USA; bernard.camins@mountsinai.org; 4Department of Graduate Medical Education, Icahn School of Medicine at Mount Sinai, New York, NY 10029, USA

**Keywords:** SARS-CoV-2, COVID-19, physician trainee, resident, fellow, risk factors

## Abstract

Occupational and non-occupational risk factors for severe acute respiratory syndrome coronavirus 2 (SARS-CoV-2) infection have been reported in healthcare workers (HCWs), but studies evaluating risk factors for infection among physician trainees are lacking. We aimed to identify sociodemographic, occupational, and community risk factors among physician trainees during the first wave of coronavirus disease 2019 (COVID-19) in New York City. In this retrospective study of 328 trainees at the Mount Sinai Health System in New York City, we administered a survey to assess risk factors for SARS-CoV-2 infection between 1 February and 30 June 2020. SARS-CoV-2 infection was determined by self-reported and laboratory-confirmed IgG antibody and reverse transcriptase-polymerase chain reaction test results. We used Bayesian generalized linear mixed effect regression to examine associations between hypothesized risk factors and infection odds. The cumulative incidence of infection was 20.1%. Assignment to medical-surgical units (OR, 2.51; 95% CI, 1.18–5.34), and training in emergency medicine, critical care, and anesthesiology (OR, 2.93; 95% CI, 1.24–6.92) were independently associated with infection. Caring for unfamiliar patient populations was protective (OR, 0.16; 95% CI, 0.03–0.73). Community factors were not statistically significantly associated with infection after adjustment for occupational factors. Our findings may inform tailored infection prevention strategies for physician trainees responding to the COVID-19 pandemic.

## 1. Introduction

New York City (NYC) was an early epicenter of coronavirus disease 2019 (COVID-19) in the United States [1]. Following identification of the first case in NYC on 1 March 2020, incident daily cases rose to a peak of 8593 cases on 10 April 2020 and gradually declined to a stable incidence of approximately 300 cases per day by June 2020 [2]. Healthcare workers (HCWs) experienced early unmitigated occupational exposure to severe acute respiratory syndrome coronavirus 2 (SARS-CoV-2) until approximately mid-March 2020, prior to implementation of standardized infection prevention protocols including universal masking, patient symptom screening, and ubiquitous telehealth, and before risk factors for transmission in healthcare settings were identified [3,4,5,6]. Reported risk factors for SARS-CoV-2 infection in HCWs include hospital department, healthcare profession, personal protective equipment (PPE) availability and use, performance of aerosol-generating procedures (AGPs), and duty hours [7]. Previously reported non-occupational factors include household and community contacts with COVID-19 cases and public transportation use [7,8].

Resident physicians and fellows (hereafter referred to as “physician trainees”) may represent a vulnerable subgroup of HCWs. On average, physician trainees work more hours per week and have fewer years of experience compared with attending physicians [9]. Additionally, evidence suggests that physician trainees are at increased risk of contracting respiratory infections, including influenza, compared with the general population [10]. Data are lacking regarding risk factors for SARS-CoV-2 infection in physician trainees. During the COVID-19 patient surge, physician trainees were assigned to work in hospital environments and perform clinical duties that may have differed from their usual training experience (hereafter referred to as “deployment”) [11,12,13]. For example, physician trainees from other training backgrounds temporarily assisted in emergency departments (ED) and intensive care units (ICU) during the COVID-19 patient surge. It is unclear whether deployment was associated with increased risk of SARS-CoV-2 infection [11,12,13,14].

Comprehensive survey-based approaches that assess both occupational and non-occupational factors are needed to understand risk factors associated with SARS-CoV-2 infection in physician trainees [15,16]. To inform and tailor existing infection prevention protocols, we aimed to assess sociodemographic, occupational, and community risk factors for SARS-CoV-2 infection among physician trainees employed by a large healthcare system in NYC during the early phases of the COVID-19 pandemic.

## 2. Materials and Methods

### 2.1. Study Setting and Design

We conducted a retrospective cohort study of physician trainees employed by the Mount Sinai Health System, comprised of eight hospitals in NYC and Long Island, NY. All active trainees from 1 January 2020 to 31 June 2020 (*n* = 2543) were eligible for this study (Figure 1). Contact information, training specialty, post-graduate year (PGY), and primary hospital training site were provided by the institution’s Office of Graduate Medical Education. Eligible trainees were invited to participate in an online survey through email, text messages, and phone calls, and were asked to retrospectively report information for the period between 1 February 2020 and 30 June 2020. The survey collected information regarding sociodemographic, occupational and community factors hypothesized to be associated with SARS-CoV-2 infection (Figure 2). Additionally, we asked physician trainees to report results of SARS-CoV-2 serum IgG antibody and reverse transcriptase-polymerase chain reaction (RT-PCR) tests. Self-reported SARS-CoV-2 test results collected from the survey were confirmed with laboratory data from Mount Sinai’s COVID-19 Employee Health Services registry. Testing was available at no cost to all employees on a voluntary and uncompensated basis. The study protocol was approved by the Institutional Review Board of Icahn School of Medicine at Mount Sinai, and written informed electronic consent was obtained from all participants.

### 2.2. Participant Enrollment

Eligible participants with valid contact information (*n* = 2354) were invited to participate through email and text message links to the electronic consent and survey on 26 June 2020. Up to five reminder invitations were sent to non-responders through 31 August 2020. In total, 328 participants who agreed to participate in this study and had available SARS-CoV-2 test results during the study period were included in the analysis.

To increase participation and to promote equitable representation of participants from all affiliated hospitals, a subset of eligible participants (*n* = 281, 11%) was selected using proportionate random sampling and stratified by hospital within the Mount Sinai Health System. Of the 281 randomly selected participants, valid contact information was available for 267 participants, of whom 72 (27%) consented to participate in the study. The response rate was higher in the randomly selected sample (27% vs. 17% in the overall sample) and was used to ascertain potential selection bias in the overall study sample.

### 2.3. Institutional Process for Employee COVID-19 Testing 

On 6 March 2020, Mount Sinai’s Employee Health Services (EHS) established an online registry for employees to voluntarily report high-risk exposures and daily symptoms of COVID-19. Healthcare providers counseled registered employees on symptom monitoring and coordinated testing and clearance for return to work. RT-PCR swabs and IgG antibody testing were available to all symptomatic employees on 7 April 2020, and to asymptomatic employees by 6 May 2020. Sensitivity and specificity of the Mount Sinai Hospital Clinical Laboratory COVID-19 ELISA antibody test is 92.5% (95% CI: 80.1–97.4%) and 100% (95% CI: 95.1–100%), respectively [17]. The sensitivity and specificity of the Roche Cobas RT-PCR test offered is 100% [18].

### 2.4. Assessment of SARS-CoV-2 Infection

We ascertained SARS-CoV-2 infection status by self-reported test results and categorized the results as positive (by IgG antibodies, RT-PCR, or both), negative (by IgG antibodies, RT-PCR, or both), or never tested. To reduce the likelihood of differential misclassification bias [19], we excluded participants who denied testing at the time of survey completion, and for whom there was no record of an IgG antibody result through 15 July 2020 in the EHS COVID-19 registry (*n* = 32). Among a subset of 199 participants who consented to review of test results, there was 100% agreement between self-reported and laboratory-confirmed results.

### 2.5. Assessment of Potential Risk Factors for SARS-CoV-2 Infection

The survey collected information regarding sociodemographic, occupational and community factors hypothesized to be associated with SARS-CoV-2 infection (Figure 2). The specific survey questions are available in Appendix A. Occupational factors included department of work during the study period, exposure to patients with confirmed or suspected (i.e., persons under investigation or PUI) SARS-CoV-2 infection, unprotected contact (without N95, eye shield, gown, or gloves) with confirmed cases or PUI, performing or attending AGPs, and factors related to deployment. Protocols for PPE use were the same for all HCWs at our institution during the study period, including physician trainees. Deployment was defined as a temporary assignment away from usual clinical duties to assist in the COVID-19 surge response, which could have required relocation to an affiliated but unfamiliar hospital within the health system, department, or change in usual patient population. For this analysis, we categorized physician trainees by specialty including: (1) primarily non-procedural specialties; (2) high-risk, primarily procedural specialties; and (3) surgical specialties (Appendix A).

Community factors assessed included primary residence (zip code), contact for more than 10 min with an individual with confirmed or suspected COVID-19 outside of work, number of adults and children in household, and primary mode of transportation to work and non-work locations.

### 2.6. Statistical Analysis

Sociodemographic, occupational and community variables were compared between groups using Fisher’s exact test for categorical variables and the Wilcoxon rank-sum test for continuous variables. Variables with a *p*-value < 0.30 in the bivariate analysis were included in Bayesian Generalized Linear Mixed Effect Regression (BGlmer) to estimate the adjusted odds of SARS-CoV-2 infection. We used BGlmer to stabilize estimates for exposure variables with zero or small numbers of observations in subgroups defined by SARS-CoV-2 infection status [20]. Using a step-by-step approach, we first tested associations in BGlmer models that were separately adjusted for sociodemographic factors (Model 1), occupational factors (Model 2), and community factors (Model 3), to evaluate confounding and reduce bias from multicollinearity and overadjustment. Variables with a *p*-value < 0.30 after backward elimination in the BGlmer model were retained in the final adjusted model. Finally, we simultaneously adjusted for sociodemographic, occupational, and community factors (Model 4) to test whether associations remained robust in a fully adjusted BGlmer model.

In addition to the BGlmer models that assessed the associations of individual factors, we used structural equation models (SEMs) to evaluate the joint associations of sociodemographic, occupational, or community factors (i.e., using latent functions) with SARS-CoV-2 infection. Three unobserved latent sociodemographic, occupational, and community functions were estimated using variables associated with SARS-CoV-2 infection in the BGlmer analysis and regressed to SARS-CoV-2 test result in the SEM. All SEMs were fitted using diagonally weighted least squares and a probit link function [21]. The root mean square error of approximation (RMSEA) for the final SEMs was < 0.05.

Sensitivity analyses included: (1) exclusion of participants with RT-PCR test results but no IgG antibody results (*n* = 314); (2) model adjustment for date of the SARS-CoV-2 test if available (*n* = 186); and (3) comparison of main characteristics between the analysis population (*n* = 328), the participants from the randomly selected sample who reported SARS-CoV-2 test results (*n* = 62), and all initially eligible participants (*n* = 2543). All statistical analyses were conducted using R version 3.6.1. Missing data for covariates (approximately 1%) were imputed using random forests with the Multivariate Imputation by Chained Equations R package [22]. The SEM analysis was conducted using the “lavaan” R package [23].

## 3. Results

### 3.1. Survey Response

Among 2354 eligible physician trainees initially contacted, 391 physician trainees (17%) responded to the invitation and 360 (15%) completed the survey (Figure 1). In total, 328 (14%) physician trainees reported having been tested for SARS-CoV-2 during the study period and were included in subsequent analysis. 

### 3.2. Participant Characteristics

Participants were of median (interquartile range) age 31 (29–33) years. Most identified as female (58% vs. 42% male), White (62% vs. 25% Asian, 8% Black and 4% other race), and non-Hispanic/Latinx (89% vs. 10% Hispanic/Latinx) (Table 1). Sixty participants (18%) reported deployment to a different hospital from their primary training site during the COVID-19 patient surge, 21% reported a change in primary clinical duties, 25% reported a department change, 12% reported greater time spent on telemedicine compared with usual clinical activities, and 10% reported a change in usual patient population (e.g., from pediatrics to adult patients).

### 3.3. SARS-CoV-2 Infection

The cumulative incidence of SARS-CoV-2 infection by 30 June 2020 was 20.1%. Of the 66 (20.1%) participants who tested positive for SARS-CoV-2 during the study period, 71% (*n* = 47) were found to be positive by IgG antibodies, 26% (*n* = 17) were found to be positive by both IgG antibodies and RT-PCR, and 3% (*n* = 2) were found to be positive by RT-PCR only (Appendix A).

### 3.4. Sociodemographic Factors and SARS-CoV-2 Infection

SARS-CoV-2 infection was more common among males (23% vs. 18% females; *p* = 0.268) and Hispanic/Latinx participants (29% vs. 19% non-Hispanic/Latinx; *p* = 0.18), and was least common among Asian participants (13% vs. 17%−27% for other races, *p* = 0.25) (Table 1). After multivariable adjustment, the odds of infection were increased among Hispanic and Latinx trainees compared with non-Hispanic or Latinx participants (fully adjusted Model 4: OR, 1.98; 95% CI, 0.72–5.46) (Table 2).

### 3.5. Occupational Factors and SARS-CoV-2 Infection

The adjusted odds of SARS-CoV-2 infection were increased for physician trainees in high-risk, primarily procedural specialties including EM, critical care, and anesthesiology (OR, 2.93; 95% CI, 1.24–6.92), and for those who reported working on inpatient medical-surgical units (OR, 2.51; 95% CI, 1.18–5.34) (Table 2). Deployment to care for unfamiliar patient populations was associated with decreased odds of infection (OR, 0.16; 95% CI, 0.03–0.73).

Assignment to work in an ED or ICU, independent of deployment, was not statistically significantly associated with infection in the bivariate analysis. Similarly, SARS-CoV-2 infection was less frequent among physician trainees who worked in ambulatory clinics and on telemedicine compared to those who reported never working in these settings, whereas infection was more likely among physician trainees who performed AGPs and who reported at least once instance of unprotected contact without N95, eye shield, gown, or gloves for over 10 min with a confirmed COVID-19 patient or PUI (Table 1). However, these associations were attenuated and not statistically significant after adjustment for other occupational factors (Table 2).

### 3.6. Community Factors and SARS-CoV-2 Infection

After multivariable adjustment for community factors (Table 2, Model 3), contact for more than 10 min with an individual with confirmed or suspected COVID-19 outside of work (OR, 2.38; 95% CI, 1.14–4.98), and use of public transit (subway or bus) as the primary mode of transportation to non-work locations (OR, 2.25; 95% CI, 1.01–5.01) were associated with increased odds for infection. Primary residence in boroughs of NYC outside of Manhattan was associated with decreased odds of infection in the bivariate analysis, however, associations of community factors with SARS-CoV-2 infection were attenuated and not statistically significant after adjustment for occupational factors (Table 2, Model 4).

### 3.7. Structural Equational Model

The SEM analysis (Table 3) produced concordant results with the multivariable adjusted regression (Table 2). The likelihood of SARS-CoV-2 infection was statistically significantly increased with an overall increase in the latent function of occupational factors. This association remained after adjustment for sociodemographic and community latent functions (adjusted SEM estimate 0.35; 95% CI, 0.15–0.54). The magnitude of the associations of sociodemographic and community factors with SARS-CoV-2 infection was attenuated and not statistically significant compared with occupational factors.

### 3.8. Sensitivity Analysis

Associations in the multivariable adjusted models remained statistically significant after excluding participants with RT-PCR results but who did not report IgG antibody results, and after adjustment for the date of the SARS-CoV-2 test (Appendix A). Physician trainees based at Mount Sinai Hospital, the largest of all affiliated sites, were overrepresented in the analysis sample (64% vs. 55% among all initially eligible participants). We did not observe additional statistically significant differences between the final analysis sample compared with eligible participants, or with the randomly selected sample (Appendix A).

## 4. Discussion

In this study of physician trainees in a large NYC-based healthcare, assignment to inpatient medical-surgical units and training in high-risk procedural specialties, including EM, anesthesiology, and critical care, were statistically significantly associated with SARS-CoV-2 infection. Assignment to unfamiliar hospital sites or clinical responsibilities was not associated with SARS-CoV-2 infection, and assignment to unfamiliar patient populations was associated with decreased risk of infection, suggesting that deployment of physician trainees was a safe strategy to respond to surging patient volume and the need for additional HCWs during the first wave of COVID-19 in NYC. Associations of community factors and SARS-CoV-2 infection were not statistically significant after adjustment for occupational factors, indicating that infection was largely attributable to occupational exposures.

In the present study of physician trainees, the cumulative incidence of SARS-CoV-2 infection by 30 June 2020 was 20.1%, similar to reported seroprevalences in other HCW subgroups and the general population of NYC during this period [24,25]. The NYC Department of Health and Mental Hygiene reported a 22.7% seroprevalence among 5101 grocery store customers tested between 19–28 April 2020, suggesting that the prevalence of SARS-CoV-2 infection among physician trainees did not exceed the frequency of infections in the general population of NYC during the initial COVID-19 wave [26]. Our results lie within the range of SARS-CoV-2 seroprevalence estimates in HCWs (1.6–36%) from studies conducted internationally during similar periods in the initial phase of the epidemic [27,28,29,30,31,32,33,34,35]. Differences observed in estimates across international studies may be, in part, due to differences in the time window of the epidemic phase examined, differences in SARS-CoV-2 testing (RT-PCR and/or IgG antibodies), differences in the PPE protocols, and mitigation strategies that varied across geographic regions and institutions, as well as the heterogenous HCW populations included in other studies.

Our findings suggest that assignment to inpatient medical-surgical units was a risk factor for SARS-CoV-2 infection, contrary to prior studies of HCWs, which found no association between department of work and infection risk [24,25]. However, previous studies did not specifically assess physician trainees, limiting comparability of previous studies with our results [24,25]. Medical-surgical units may have been less familiar to participants who, prior to the COVID-19 pandemic, spent a greater proportion of duty hours at ambulatory care sites, or in operating rooms or procedural environments. Among physician trainees from surgical and primary care specialties, decreased familiarity with routine infection prevention protocols specific to medical-surgical units may further explain our findings. Additionally, caring for PUI in medical-surgical units may have diminished the urgency of adherence to optimal infection prevention protocols, compared with caring for confirmed COVID-19 patients. Finally, working in an ED or ICU was not associated with SARS-CoV-2 infection in this study, consistent with prior reports of HCWs in the greater New York area [24,25].

Physician trainees in high-risk procedural specialties were at increased risk for SARS-CoV-2 infection in this study, consistent with prior studies. Breazzano et al. reported a higher frequency of SARS-CoV-2 infections among EM and anesthesiology residents compared with other specialties [11]. EM, anesthesiology, and critical care physicians in training routinely perform endotracheal intubation, and likely had unmitigated exposure to aerosolized virus from undiagnosed COVID-19 patients early in the study period, prior to implementation of routine infection prevention protocols [36,37,38]. Taken in context with prior evidence, our findings suggest that identifying COVID-19-positive patients prior to performing intubation and other AGPs, as well as the use of PPE, contribute to reducing the risk of SARS-CoV-2 infection in physician trainees.

Deployment to unfamiliar hospital sites and clinical responsibilities was not a statistically significant risk factor for SARS-CoV-2 infection, despite limited time for patient surge planning [15,16]. Moreover, we found that deployment to care for unfamiliar patient populations was associated with decreased adjusted odds of infection. Among survey respondents, pediatrics residents and fellows most frequently reported a patient population change, most commonly to care for adult patients in ED or ICU environments. Deployment strategies differed according to department in the Mount Sinai Health System, and the Department of Pediatrics and Mount Sinai Hospital deployed physician trainees on a voluntary basis. It is plausible that trainees who cared for unfamiliar patient populations may have performed more administrative tasks and had fewer instances of direct patient care, thus reducing direct exposures and SARS-CoV-2 transmission risk.

Use of public transportation, particularly use of the subway or bus, was associated with increased risk of SARS-CoV-2 infection in our study prior to adjustment for occupational factors, in agreement with findings from a previous neighborhood-level study in NYC [39] and two prior reports of HCWs [11,40]. However, the associations of community factors with infection risk were attenuated and non-significant after adjustment for occupational factors in our study. Our findings suggest that community exposure, defined by area of residence, use of transportation, and direct contact with an individual with suspected or confirmed COVID-19 outside of work, may contribute to infection risk among NYC-based physician trainees, albeit less significantly than occupational exposures.

Strengths of our study include the collection of robust data directly from physician trainees pertaining to both occupational and community exposures in NYC, an early epicenter of COVID-19 in the U.S. Associations of occupational factors and SARS-CoV-2 infection are strengthened by our ability to verify self-reported test results with laboratory-confirmed data for most participants. Results from the sensitivity analysis indicated similar sociodemographic characteristics among all eligible participants, the randomly selected subset, and study participants in the analysis sample, reducing the likelihood of selection bias in our study. The high sensitivities and specificities of RT-PCR and antibody tests offered at our institution during the study period reduce the likelihood of SARS-CoV-2 infection misclassification in study participants. However, we cannot rule out the potential measurement error of bias due to participants whose self-reported test results could not be verified with data from the EHS registry, and from tests with different sensitivities and specificities that may have been performed outside of the Mount Sinai Health System. Finally, our results may underestimate the cumulative incidence of SARS-CoV-2 infection, as some physician trainees could have been infected but asymptomatic, and therefore not tested during the study period. Finally, results may not be generalizable to physician trainees outside of NYC, as hospital infection prevention protocols and community transmission vary by geographic location.

## 5. Conclusions

Among physician trainees at a large healthcare system situated in an early U.S. epicenter of COVID-19, assignment to medical-surgical units and training in high-risk procedural specialties were most robustly associated with SARS-CoV-2 infection, out of a comprehensive list of occupational, community, and sociodemographic factors assessed. Our findings further suggest that deployment of physician trainees to non-routine hospital sites and clinical responsibilities was a strategy to respond to surging patient volume during the initial phases of the COVID-19 pandemic and may be safe during current international patient surges. Community exposures (e.g., contact with COVID-19 cases outside of the working environment and public transit use) may have also contributed to SARS-CoV-2 infection in NYC physician trainees, however these associations were significantly attenuated in this study by adjustment for occupational factors, indicating that infection in physician trainees was largely attributable to occupational exposures. In summary, our findings can inform more tailored infection prevention strategies for physician trainees during the COVID-19 pandemic.

## Figures and Tables

**Figure 1 ijerph-18-05274-f001:**
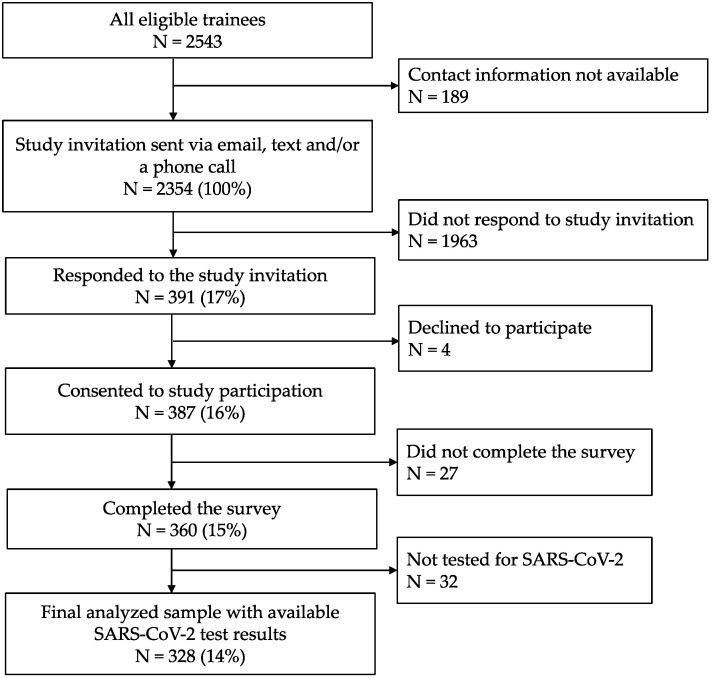
Flow chart of participant recruitment and survey responses.

**Figure 2 ijerph-18-05274-f002:**
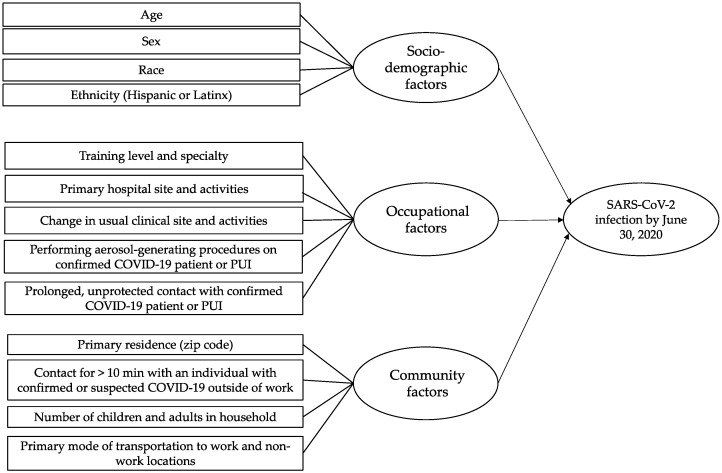
Risk factors hypothesized to be associated with SARS-CoV-2 infection in physician trainees.

**Table 1 ijerph-18-05274-t001:** Sociodemographic, occupational and community risk factors by SARS-CoV-2 test status.

Variable	Negative SARS-CoV-2 Test (*n* = 262)	Positive SARS-CoV-2 Test (*n* = 66)	*p*-Value
**Sociodemographic factors**			
**Age, years, median (IQR)**	31 (29–33)	30 (28–33)	0.36
**Sex, no. (%)**			0.27
Female	155 (82)	34 (18)	
Male	107 (77)	32 (23)	
**Race, no. (%)**			0.25
White	156 (77)	46 (23)	
Asian	71 (87)	11 (13)	
Black	19 (73)	7 (27)	
Other	10 (83)	2 (17)	
Missing	6	0	
**Hispanic/Latinx, no. (%)**			0.18
No	237 (81)	56 (19)	
Yes	24 (71)	10 (29)	
Missing	1	0	
**Occupational factors**			
**Training specialty, no. (%)**			0.002
Hospital-based, primarily non-procedural	180 (85)	33 (15)	
High-risk procedural	32 (62)	20 (38)	
Surgical	41 (77)	12 (23)	
Missing	9	1	
**PGY level, no. (%)**			0.57
1	55 (75)	18 (25)	
2	51 (82)	11 (18)	
≥3	156 (81)	37 (19)	
**Resident or fellowship, no. (%)**			0.88
Fellowship	69 (81)	16 (19)	
Residency	193 (79)	50 (21)	
**Primary hospital site, no. (%)**			0.27
Beth Israel Medical Center	23 (82)	5 (18)	
Elmhurst Hospital Center	15 (100)	0 (0)	
Institute for Family Health	4 (67)	2 (33)	
Mount Sinai Hospital	166 (79)	45 (21)	
North Central Bronx	1 (100)	0 (0)	
Queens Hospital Center	6 (86)	1 (14)	
South Nassau Communities Hospital	2 (50)	2 (50)	
St. Luke’s Roosevelt Hospital	45 (80)	11 (20)	
***Occupational setting***			
**Medical-surgical unit, no. (%)**			0.24
No	89 (84)	17 (16)	
Yes	173 (78)	49 (22)	
**Emergency department, no. (%)**			0.64
No	194 (80)	47 (20)	
Yes	68 (78)	19 (22)	
**ICU, no. (%)**			>0.99
No	154 (80)	39 (20)	
Yes	108 (80)	27 (20)	
**Ambulatory clinic, no. (%)**			0.04
No	174 (77)	53 (23)	
Yes	88 (87)	13 (13)	
**Telemedicine, no. (%)**			0.047
No	181 (77)	54 (23)	
Yes	81 (87)	12 (13)	
***High-risk occupational exposures***			
**Direct care for confirmed COVID-19 case or PUI, no. (%)**			0.29
No	33 (87)	5 (13)	
Yes	229 (79)	61 (21)	
**Performed or attended an AGP on confirmed COVID-19 case or PUI, no. (%)**			0.05
No	127 (85)	23 (15)	
Yes	134 (76)	43 (24)	
Missing	1	0	
**Contact > 10 mins with confirmed *without N95* COVID-19 case or PUI, no. (%)**			0.07
No	182 (83)	37 (17)	
Once	42 (76)	13 (24)	
Twice or more	36 (69)	16 (31)	
Missing	2	0	
**Contact > 10 mins *without eye protection* with confirmed COVID-19 case or PUI, no. (%)**			0.09
No	155 (83)	31 (17)	
Once	44 (80)	11 (20)	
Twice or more	61 (72)	24 (28)	
Missing	2	0	
**Contact > 10 mins *without gown* with confirmed COVID-19 case or PUI, no. (%)**			0.01
No	174 (84)	32 (16)	
Once	37 (77)	11 (23)	
Twice or more	48 (68)	23 (32)	
Missing	3	0	
**Contact > 10 mins *without gloves* with confirmed COVID-19 case or PUI, no. (%)**			0.12
None	225 (81)	52 (19)	
Once or more	34 (71)	14 (29)	
Missing	3	0	
***Deployment factors***			
**Change in usual hospital, no. (%)**			0.59
No	212 (79)	56 (21)	
Yes	50 (83)	10 (17)	
**Change in usual clinical activities, no. (%)**			0.87
No	206 (80)	53 (20)	
Yes	56 (81)	13 (19)	
**Change in usual patient population, no. (%)**			<0.001
No	230 (78)	66 (22)	
Yes	32 (100)	0 (0)	
**Change in usual department, no. (%)**			0.34
No	193 (78)	53 (22)	
Yes	69 (84)	13 (16)	
**More time on telemedicine than usual, no. (%)**			0.05
No	226 (78)	63 (22)	
Yes	36 (92)	3 (8)	
**Community factors**			
**Primary residence, no. (%)**			0.06
Manhattan	202 (77)	60 (23)	
Queens	28 (93)	2 (7)	
Brooklyn	12 (100)	0 (0)	
Bronx	5 (100)	0 (0)	
Outside of NYC	13 (76)	4 (24)	
Missing	2	0	
**Contact > 10 mins with individual confirmed or suspected COVID-19 outside of work, no. (%)**			0.008
No	212 (83)	43 (17)	
Yes	50 (68)	23 (32)	
**Number of adults in household, no. (%)**			0.64
1 (self)	72 (82)	16 (18)	
≥ 2	189 (79)	50 (21)	
Missing	1	0	
**Number of children in household, no. (%)**			0.19
0	214 (78)	59 (22)	
≥ 1	46 (87)	7 (13)	
Missing	2	0	
***Primary mode of transportation to work***			
**Public transit (subway or bus), no. (%)**			0.32
No	165 (82)	37 (18)	
Yes	97 (77)	29 (23)	
**Cab or rideshare, no. (%)**			0.37
No	183 (81)	42 (19)	
Yes	79 (77)	24 (23)	
**Private vehicle, bicycle or walking, no. (%)**			0.86
No	53 (82)	12 (18)	
Yes	209 (79)	54 (21)	
***Primary mode of transportation to non-work location***			
**Public transit (subway or bus), no. (%)**			0.07
No	220 (82)	49 (18)	
Yes	42 (71)	17 (29)	
**Cab or rideshare, no. (%)**			0.049
No	220 (82)	48 (18)	
Yes	42 (70)	18 (30)	
**Private vehicle, bicycle or walking, no. (%)**			0.08
No	12 (63)	7 (37)	
Yes	250 (81)	59 (19)	

Abbreviations: IQR, interquartile range; PGY, post-graduate year; PUI, patient under investigation (suspected to be positive for SARS-CoV-2 and pending laboratory result); ICU, intensive care unit; AGP, aerosol-generating procedure.

**Table 2 ijerph-18-05274-t002:** Adjusted effect estimates for associations of sociodemographic, occupational and community factors with SARS-CoV-2 infection.

Variable	Model 1: Sociodemographic Factors		Model 2: Occupational Factors		Model 3: Community Factors		Model 4: Final Adjusted Model	
	OR	95% CI	OR	95% CI	OR	95% CI	OR	95% CI
Race								
White (ref)	1.00	-					1.00	-
Asian	0.53	0.23, 1.24					0.53	0.24, 1.15
Black	1.34	0.45, 3.98					1.42	0.50, 4.01
Other	0.43	0.08, 2.47					0.64	0.14, 2.92
Hispanic/Latinx								
No (ref)	1.00	-					1.00	-
Yes	2.18	0.73, 6.47					1.98	0.72, 5.46
Change in usual patient population								
No (ref)			1.00	-			1.00	-
Yes			0.09	0.01, 0.67			0.16	0.03, 0.73
Medical/surgical unit								
No (ref)			1.00	-			1.00	-
Yes			2.96	1.27, 6.91			2.51	1.18, 5.34
Ambulatory clinic								
No (ref)			1.00	-			1.00	-
Yes			0.53	0.24, 1.17			0.61	0.29, 1.30
Contact >10 mins *without N95* with confirmed COVID-19 case								
Never (ref)			1.00	-			1.00	-
Once			1.47	0.62, 3.48			1.24	0.55, 2.75
Twice or more			1.72	0.75, 3.94			1.59	0.74, 3.43
Training specialty								
Hospital-based, primarily non-procedural (ref)			1.00	-			1.00	-
High-risk procedural			4.29	1.62, 11.33			2.93	1.24, 6.92
Surgical			1.98	0.81, 4.89			1.51	0.65, 3.50
Number of children in household								
0 (ref)					1.00	-	1.00	-
≥ 1					0.52	0.20, 1.38	0.59	0.23, 1.48
Contact > 10 mins with individual confirmed or suspected COVID-19 outside of work								
No (ref)					1.00	-	1.00	-
Yes					2.38	1.14, 4.98	1.58	0.78, 3.17
Primary mode of transportation to location other than work: *public transit (subway or bus)*								
No (ref)					1.00	-	1.00	-
Yes					2.25	1.01, 5.01	1.85	0.85, 3.99
Primary mode of transportation to location other than work: *private vehicle, bicycle, walking*								
No (ref)					1.00	-	1.00	-
Yes					0.44	0.14, 1.40	0.42	0.14, 1.27
Primary residence (zip code)								
Manhattan (ref)					1.00	-	1.00	-
Queens					0.24	0.06, 0.94	0.34	0.10, 1.20
Brooklyn					0.21	0.03, 1.64	0.30	0.06, 1.62
Bronx					0.40	0.04, 3.98	0.48	0.08, 3.08
Outside of NYC					1.48	0.40, 5.49	1.51	0.44, 5.20

Abbreviations: OR, odds ratio; CI, confidence interval; ref, reference.

**Table 3 ijerph-18-05274-t003:** Adjusted effect estimates for associations of sociodemographic, occupational and community latent functions with SARS-CoV-2 infection.

Exposure Latent Functions	SEM 1 ^a^		SEM 2 ^b^		SEM 3 ^c^		SEM 4 ^d^	
	OR	95% CI	OR	95% CI	OR	95% CI	OR	95% CI
Sociodemographic factors	0.09	−0.07, 0.25					0.13	−0.06, 0.31
Occupational factors			0.33	0.13, 0.53			0.35	0.15, 0.54
Community factors					0.12	−0.08, 0.32	0.10	−0.12, 0.33

Abbreviations: SEM, structural equation model; OR, odds ratio; CI, confidence interval. ^a^ SEM 1adjusted for the latent function of sociodemographic factors. ^b^ SEM 2: adjusted for the latent function of occupational factors. ^c^ SEM 3: adjusted for the latent function of community factors. ^d^ SEM 4: simultaneously adjusted for all latent functions.

## Data Availability

The data presented in this study are available by request from the corresponding author. The data are not publicly available due to privacy restrictions.

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
