# Peer review of "Factors Associated with SARS-CoV-2 Infection in Physician Trainees in New York City during the First COVID-19 Wave"

_ijerph, 2021, doi:10.3390/ijerph18105274_

Round 1

Reviewer 1 Report

With interest, I have read the ms. by Pawloski and coll, titled “Factors Associated with SARS-CoV-2 Infection in Resident Physicians and Fellows in New York City During the First COVID-19 Wave”. The paper is well written and is noteworthy in that it allows to evaluate occupational/community exposure to SARS-CoV-2 in a sample of medical residents.

Other studies conducted in healthcare workers revealed high variation of SARS-CoV-2-ab seroprevalence and this study contributes to refine these estimates. Within this context, the paper may benefit from further comparison with similar studies (not only conducted in the same geographical area – like the study of Moscola et al., ref #23), pointing out differences and similarities of other sero-epidemiological surveys, also taking into account the inclusion of PCR results. For example, a recently published paper on occupational exposure to SARS-CoV-2 also antibody prevalence in subjects with close contacts to public or major risk exposure (such as healthcare workers) - doi: 10.3390/ijerph18052567

Or other studies that added swab testing, e.g., doi: 10.1016/S1473-3099(20)30589-2

Minor comments.

Line 23 - SARS-CoV-2

Line 74 – it is not clear what results of serum IgG antibody and swabs have been “self-reported”.

Line

Line 87-91 – In my opinion, these are results.

Were there differences protection measures between the study participants or “normal” healthcare workers? I mean, for example, if HCWs were provided with personal protective equipment (mask, eye shield, etc) and trainees weren’t…

Author Response

Response to Reviewers

May 11, 2021

Ref. IJERPH-1200960: Factors Associated with SARS-CoV-2 Infection in Resident Physicians and Fellows in New York City During the First COVID-19 Wave

Dear Prof. Dr. Tchounwou,

We are grateful to the Reviewers for their thoughtful review and comments on our manuscript. Please find our point-by-point response to the comments provided by the Reviewers. Please feel free to contact us if you have more queries or questions. Thank you.

Reviewer 1:

With interest, I have read the Ms. by Pawloski and coll, titled “Factors Associated with SARS-CoV-2 Infection in Resident Physicians and Fellows in New York City During the First COVID-19 Wave”. The paper is well written and is noteworthy in that it allows to evaluate occupational/community exposure to SARS-CoV-2 in a sample of medical residents.

Other studies conducted in healthcare workers revealed high variation of SARS-CoV-2-ab seroprevalence and this study contributes to refine these estimates. Within this context, the paper may benefit from further comparison with similar studies (not only conducted in the same geographical area – like the study of Moscola et al., ref #23), pointing out differences and similarities of other sero-epidemiological surveys, also taking into account the inclusion of PCR results. For example, a recently published paper on occupational exposure to SARS-CoV-2 also antibody prevalence in subjects with close contacts to public or major risk exposure (such as healthcare workers) - doi: 10.3390/ijerph18052567

Or other studies that added swab testing, e.g., doi: 10.1016/S1473-3099(20)30589-2

RESPONSE:

Thank you for your thoughtful review and comments on our manuscript. We appreciate your recommendation to include further comparison with similar studies conducted in diverse geographical areas. There are notables differences in the seroprevalences reported for health care workers across studies which may be explained in part by differences in the time window of the epidemic phase examined, differences in SARS-CoV-2 testing (RT-PCR and/or IgG antibodies), differences in the PPE protocols and mitigation strategies that varied across geographic regions and institutions, as well as the heterogenous HCW populations in other studies compared with ours that focused exclusively on physician trainees. We have now included estimates from international seroprevalence studies among healthcare workers during similar periods in the initial phase of the COVID-19 pandemic and commented on these issues in our discussion (see additional references 27-35, and lines 280-287 of the revised manuscript).

Minor comments:

Line 23 - SARS-CoV-2

RESPONSE:

Thank you for noting this error. We have corrected it to SARS-CoV-2 in line 21 of the revised manuscript.

Line 74 – it is not clear what results of serum IgG antibody and swabs have been “self-reported”.

RESPONSE:

In the survey, we asked trainees to report whether they had been tested for SARS-CoV-2 by RT-PCR or serum IgG antibody tests during the study period. The self-reported SARS-COV-2 results collected from the survey were confirmed with laboratory data extracted from Mount Sinai’s COVID-19 Employee Health Services registry, and the level of agreement was 100%. We have rephrased the text to clarify that we were referring to the self-reported SARS-COV-2 test results collected from the survey in lines 77-81 of the revised manuscript.

Line 87-91 – In my opinion, these are results.

RESPONSE:

Thank you for this suggestion. Data regarding the survey response is now included in the Result section 3.1 (lines 179-183 of the revised manuscript).

Were there differences protection measures between the study participants or “normal” healthcare workers? I mean, for example, if HCWs were provided with personal protective equipment (mask, eye shield, etc) and trainees weren’t…

RESPONSE:

There were no differences in personal protection equipment protocols between the study participants and other healthcare workers employed by our institution during the study period. We have now clarified this in lines 134-136 of the revised manuscript.

Reviewer 2 Report

Overall comment.

A good epidemiological analysis, well written and structured.

Some aspects to improve: especially clarifying the aims and value of the analyses conducted (either for domestic and international contexts).

Some specific comments below (questions are intended to illustrate sections of the manuscript that need further clarification):

Abstract

The abstract focuses on the methods and especially the results. It is not clear what the justification of the study is and why some factors (i.e. community factors) were relevant for the conclusions.

Introduction

Line 48. It is not clear the link between the reference to “previous factors reported” and the study cited in [8]. -this requires further clarification (consider linking it to the first sentence in the next paragraph where you cite the study again).

Line 54. “trainees” better to be referred to as “medical trainees”.

Line 54-55. What do you mean by “unfamiliar clinical roles”?

Line 57. It will help to understand the approach you followed if you further expand on the “comprehensive approaches needed” you refer at the start of this sentence.

The aim of the study is not very clear from the last sentence of the introduction. What was your ultimate aim in assessing sociodemographic, occupational and community risk factors for SARS-CoV-2 infection among medical trainees?

Methods

A mention of the ethics clearance for the study is needed.

Lines 104-107. Further clarification on why these details are important in this study

Line 123. In the parenthesis, you meant “and” gloves instead of “or”?

Line 141. What specified subgroups you included?

It would help to add further clarification on why running a full model (4) and the models separated for each risk factor categories (models 1-3)

Lines 156-160. Maybe these are better located after the introduction of the BGLmer models.

I could not get the reason to run the SEM models and how this adds to the GLmer models?

Results

Line 166. Spell “IQR” in full.

Discussion

Line 247. Was the deployment intended as a mitigation strategy?

Line 251. Did you mean the cumul. Incidence in the study population?

Lines 258-260. Sentence is not clear. Also is this considering the different populations included in these studies?

Lines 261-263. Has this been previously mentioned in the manuscript? It seems to be relevant to the results.

Parag, lines 270-276. What are the implications of these consistencies between studies?

Lines 290-293 seem to contradict a bit lines 248-250.

Lines 301-304 and 304-305. What are the potential implications of these limitations?

Suggest focusing on the limitations rather than the strengths (although the strengths commented in the manuscript seem to be relevant to how some of the limitations were addressed).   

Conclusions

The conclusions can be more robust and meaningful of the implications of this research (reads more like a summary of the results).

Author Response

Response to Reviewers

May 11, 2021

Ref. IJERPH-1200960: Factors Associated with SARS-CoV-2 Infection in Resident Physicians and Fellows in New York City During the First COVID-19 Wave

Dear Prof. Dr. Tchounwou,

We are grateful to the Reviewers for their thoughtful review and comments on our manuscript. Please find our point-by-point response to the comments provided by the Reviewers. Please feel free to contact us if you have more queries or questions. Thank you.

Reviewer 2:

Overall comment.

A good epidemiological analysis, well written and structured.

Some aspects to improve: especially clarifying the aims and value of the analyses conducted (either for domestic and international contexts).

RESPONSE:

Thank you for your review and thoughtful comments on our manuscript. We have now revised the sections of abstract, introduction and discussion to clarify the study aims and the value of our findings.

Some specific comments below (questions are intended to illustrate sections of the manuscript that need further clarification):

Abstract

The abstract focuses on the methods and especially the results. It is not clear what the justification of the study is and why some factors (i.e. community factors) were relevant for the conclusions.

RESPONSE:

We investigated both occupational and community risk factors for SARS-CoV-2 infection in physician trainees, based upon prior evidence that exposures in both settings may contribute to infection risk among broader populations of healthcare workers. We have now elaborated upon our study justification in lines 15-17 of the revised manuscript.

Introduction

Line 48. It is not clear the link between the reference to “previous factors reported” and the study cited in [8]. -this requires further clarification (consider linking it to the first sentence in the next paragraph where you cite the study again).

RESPONSE:

 Thank you for noting this discrepancy. The study by Staiger et al (previously reference 8) is now correctly cited as reference 9 in the next paragraph (line 52 of the revised manuscript). We have included an additional study by Baker et al (new reference 8) that identified community contact with COVID-19 cases as a risk factor for SARS-CoV-2 in HCWs.

Line 54. “trainees” better to be referred to as “medical trainees”.

RESPONSE:

 Thank you for this helpful suggestion. As many participants were trainees in surgical specialties, we have chosen to describe participants as “physician trainees.” We have added this descriptor throughout the body of the manuscript.

Line 54-55. What do you mean by “unfamiliar clinical roles”?

RESPONSE:

 During the first wave of COVID-19 in NYC, physician trainees were assigned to clinical roles and responsibilities that may have been different from their usual training experience. For example, physician trainees from other training backgrounds temporarily assisted in emergency departments and intensive care units during the COVID-19 patient surge, which could have been a risk factor for SARS-CoV-2 infection. We have now clarified this in lines 54-59 of the revised manuscript.

Line 57. It will help to understand the approach you followed if you further expand on the “comprehensive approaches needed” you refer at the start of this sentence.

RESPONSE:

Thank you for this suggestion. We have clarified that we used a comprehensive survey-based approach to study both occupational and non-occupational risk factors potentially relevant for SARS-CoV-2 infection risk in physician trainees in line 60 of the revised manuscript.

The aim of the study is not very clear from the last sentence of the introduction. What was your ultimate aim in assessing sociodemographic, occupational and community risk factors for SARS-CoV-2 infection among medical trainees?

RESPONSE:

Thank you for this helpful suggestion. We have elaborated on our aim to inform existing infection prevention protocols for physician trainees in lines 63-65 of the revised manuscript.

Methods

A mention of the ethics clearance for the study is needed.

RESPONSE:

This study was approved by the Institutional Review Board of Icahn School of Medicine at Mount Sinai. We now more clearly state this in Methods, lines 82-84 of the revised manuscript. The IJERPH journal also requires a “Institutional Review Board Statement” from authors which can be found after the “Funding” section.

Lines 104-107. Further clarification on why these details are important in this study

RESPONSE:

The high sensitivities and specificities of the RT-PCR and IgG antibody tests offered at our institution reduce the potential for measurement error and misclassification of SARS-CoV-2 infection occurrence in our study. We believe this is essential information to report in the Methods as it will facilitate readers’ interpretation and comparison of our results with those from similar studies. We described this strength and residual limitations lines 339-344 of the revised manuscript.

Line 123. In the parenthesis, you meant “and” gloves instead of “or”?

RESPONSE:

We assessed risk of infection according to use of each personal protective item individually. The specific separate survey questions referring to personal protective equipment are shown in Supplemental Table 1.

Line 141. What specified subgroups you included?

RESPONSE:

We referred to subgroups defined by exposure variables and by SARS-CoV-2 infection status. We used BGlmer to stabilize estimates where there is sparsity in subgroups, i.e., zero or small numbers of observations in subgroups defined by SARS-CoV-2 infection status and exposure variables [20]. Exposure variables with zero or small numbers of observations in subgroups defined by SARS-CoV-2 infection status are found in Table 1.

It would help to add further clarification on why running a full model (4) and the models separated for each risk factor categories (models 1-3)

RESPONSE:

We used a step-by-step approach to assess relationships between categories of risk factors and SARS-CoV-2 infection to clearly illustrate confounding and reduce bias from multicollinearity and overadjustment. We have now clarified this in the text (lines 151-156). For example, model 3 demonstrates that community factors (contact with COVID-19 case outside of work, use of subway or bus) are statistically significantly associated with infection, however after adjustment for occupational factors in model 4, associations of community factors were attenuated and no longer significant. We present results of models 1-4 in the manuscript, as this can inform the reader about confounding and whether results remained consistent (or not) in the fully adjusted model. This facilitates the interpretability of our findings and allows a more direct comparison of our results with other studies that had available data and were able to adjust only for occupational factors.

Lines 156-160. Maybe these are better located after the introduction of the BGLmer models.

I could not get the reason to run the SEM models and how this adds to the GLmer models?

RESPONSE:

BGlmer models assess the associations of individual risk factors with SARS-CoV-2 infection, while the SEM models assess the joint associations of sociodemographic, occupational or community factors using latent functions (i.e., an estimate of the intersection of multiple risk factors) with SARS-CoV-2 infection. We have now clarified this in lines 161-164 in the revised manuscript.

Results

Line 166. Spell “IQR” in full.

RESPONSE:

We have made this correction in line 186 of the revised manuscript.

Discussion

Line 247. Was the deployment intended as a mitigation strategy?

RESPONSE:

Thank you for highlighting this. We have now clarified this point. In response to the surging COVID-19 patient volume, trainees were deployed to sites where additional HCW were needed due to increases in case volume and HCW shortages resulting from illness. More accurately stated, our results suggest that deployment was a safe strategy to respond to surging COVID-19 patient volume. We have now changed this phrasing in lines 269-270 and 355-358 of the revised manuscript.

Line 251. Did you mean the cumul. Incidence in the study population?

RESPONSE:

Yes. We have now rephrased this sentence for clarity in line 274 of the revised manuscript.

Lines 258-260. Sentence is not clear. Also is this considering the different populations included in these studies?

RESPONSE:

Thank you for highlighting this point. Our results suggest that assignment to inpatient medical-surgical units was a risk factor for SARS-CoV-2 infection contrary to few prior studies of HCWs, which found no association between department of work and infection risk. However, previous studies did not specifically assess physician traineeslimiting comparability of previous studies with our results. We have now clarified this in lines 288-292 in the revised manuscript.

Lines 261-263. Has this been previously mentioned in the manuscript? It seems to be relevant to the results.

RESPONSE:

We described this rationale in the discussion as a potential explanation for the finding that assignment to medical-surgical units was an independent risk factor for infection. We did not specifically assess whether medical-surgical units were familiar to trainees assigned to this area, so it was not included in the results section.

Parag, lines 270-276. What are the implications of these consistencies between studies?

RESPONSE:

Thank you for highlighting this point. Taken in context with prior evidence, our findings suggest that identifying COVID-19-positive patients before intubation and other AGPs, as well as the use of personal protective equipment, contribute to reducing the risk of SARS-CoV-2 infection. We have included this statement in lines 308-311 of the revised manuscript.

Lines 290-293 seem to contradict a bit lines 248-250.

RESPONSE:

We have rephrased this paragraph to clarify that use of public transportation, particularly use of the subway or bus, was associated with increased risk of SARS-CoV-2 infection to adjustment for occupational factors, suggesting that community exposures may contribute to infection risk among NYC-based physician trainees, albeit less significantly than occupational exposures. We clarified this point in lines 323-331 of the revised manuscript.

Lines 301-304 and 304-305. What are the potential implications of these limitations?

RESPONSE:

 We have revised this paragraph to clarify that the high sensitivities and specificities of the RT-PCR and antibody tests offered at our institution during the study period reduce the likelihood of SARS-CoV-2 infection misclassification in study participants. However, we cannot rule out potential measurement error or bias due participants where self-reported test results could not be verified with data from the EHS registry and tests with different sensitivities and specificities may have been performed outside of the Mount Sinai Health System. Finally, our results may underestimate the cumulative incidence of SARS-CoV-2 infection as some physician trainees could have been infected but asymptomatic and therefore not tested by the time of this study. Finally, results may not be generalizable to physician trainees outside of NYC, as hospital infection prevention protocols and community transmission vary by geographic location.  We have included this information in lines 339-349 of the revised manuscript.

Suggest focusing on the limitations rather than the strengths (although the strengths commented in the manuscript seem to be relevant to how some of the limitations were addressed).

RESPONSE:

Thank you for this suggestion. Based on the above comments, we have revised the discussion of strengths and limitations in this paragraph, described in the previous response.    

Conclusions

The conclusions can be more robust and meaningful of the implications of this research (reads more like a summary of the results).

RESPONSE:

We have revised the paragraph of conclusions accordingly, in lines 355-364 of the revised manuscript.